# Thermochromic Polymeric Films for Applications in Active Intelligent Packaging—An Overview

**DOI:** 10.3390/mi12101193

**Published:** 2021-09-30

**Authors:** Airefetalo Sadoh, Samiha Hossain, Nuggehalli M. Ravindra

**Affiliations:** Interdisciplinary Program in Materials Science & Engineering, New Jersey Institute of Technology, Newark, NJ 07102, USA; aas58@njit.edu (A.S.); sh348@njit.edu (S.H.)

**Keywords:** thermochromism, polymers, intelligent packaging, thermal sensor

## Abstract

The need for passive sensors to monitor changes in temperature has been critical in several packaging related applications. Most of these applications involve the use of bar codes, inks and equipment that involve constant complex electronic manipulation. The objective of this paper is to explore solutions to temperature measurements that not only provide product information but also the condition of the product in real time, specifically shelf-life. The study will explore previously proposed solutions as well as plans for modified approaches that involve the use of smart polymers as temperature sensors.

## 1. Introduction

For the past decade, the scientific community has invested heavily into researching and identifying ways to map, monitor and control temperature in real-time due to the considerable influence temperature has on multiple aspects of our daily lives such as the life expectancy of dairy, food and pharmaceuticals and in the manufacture, storage and transport of these products [1,2]. Another critical example for the need to monitor and control temperature is in the storage, distribution and transportation of drugs, vaccines and biologics [3]. It has been reported that three of the COVID-19 vaccines that are most developed, i.e., Moderna, BioNTech and Pfizer need to be stored at extremely cold temperatures. Moderna’s COVID-19 candidate requires a storage temperature of −20 °C while Pfizer’s vaccine requires ultra-cold temperatures, as cold as −70 °C [4,5]. If these vaccines are exposed to temperatures that deviate even slightly from their ideal conditions, they may spoil and become unusable. Most vaccines, including the recently CDC approved Janssen COVID-19 vaccines by Johnson & Johnson, are usually stored between 2.22 °C and 7.78 °C. With the high demand for temperature measurements across multiple industries, various techniques and materials need to be considered to create sensors that are suitable for such diverse applications. One such technique that is examined in this review is the use of “smart” polymers in a sensing system to study, control and monitor the changes in temperature [6].

Historically, the research group’s focus, at NJIT, has been on inorganic temperature measurement systems such as vanadium oxide-based microbolometers, amorphous silicon microbolometers, HgCdTe infrared detectors and silicides such as PtSi on silicon & RhSi on silicon Schottky barrier detectors. The shift in focus has been largely due to the changing circumstances surrounding biologics transport specifically regarding COVID 19 vaccine. We have recently moved towards 3D printable devices and thus we are currently focusing on polymer-based temperature sensors. The end goal of the study is to create a 3D printable polymer temperature sensor that allows us to instantaneously visually assess the temperature range of a system and has the potential to be commercially viable. 

Smart materials are substances that react to a physical or chemical stimulus in its external environment with a quantifiable and reproducible response such as a change in material properties [6]. One such phenomenon is chromism, which is the ability of a substance to undergo color transition in response to a chemical or physical stimulus in its environment such as temperature, pH, humidity, pressure, etc. Smart polymers are increasingly being investigated in the last several decades. Polymers are adaptable to many applications due to their wide range of tunable chemical and physical properties which include structural stability, biocompatibility and ease of processing which makes them excellent candidates for subsequent integration with detection devices [6].

Thermochromism is the color change observed in chromic materials due to temperature fluctuations. Thermochromic systems are either direct or indirect. Direct systems change color with temperature differences in the environment, but indirect systems change colors when the temperature of the medium itself is affected. Stereoisomerism, liquid crystals, and molecular rearrangements are all examples of direct thermochromic systems while indirect systems need a combination of elements such as a leuco dye, a developer, and a solvent to form a functioning system [7]. Direct systems are advantageous as they are not dependent on additives for color transitions, which could be potentially toxic, and thus have a much more versatile field of applications. Temperature induced conformational changes within the polymer chain are responsible for the observed color changes due to causing a shift in the energy between the frontier orbitals. Changes in the structure lead to an increase in the energy gap, shorter conjugation lengths and wavelength shift in optical spectra [8].

Thermochromic materials have been used extensively in temperature sensing systems across various fields such as food packaging, medical thermography, the non-destructive testing of engineered articles, electronic circuitry, and pharmaceutical industry [7]. From a historic perspective, this year marks the 140th Birthday of Hermann Staudinger, the Father of Polymer Chemistry. This year is also the 101st anniversary of the proposal of the concept of macromolecules, that was characterized as polymers, by Staudinger in 1920 and the discovery of covalent bonds [9].

Sensors in general are often reflective of these diverse applications as scientists often need to control complex systems [10]. Figure 1 showcases the key milestones and progression of wearable sensors during the last 40 years. The first wearable sensor was a chest strap developed in 1980. This strap highlighted the importance of using flexible materials (such as polymers) as a substrate for the fabrication of sensors [11]. Polymers play an extremely vital role in these systems because they possess a variety of characteristics that enable them to adapt and meet the demand for a wide range of applications [12]. Essentially, polymeric materials can be tailored to respond in a favorable manner for whatever signal is required for control in sensing systems. For example, in health care, they can be used to mimic the natural behavior of sense organs (contact lenses), control of chemical reaction (modified polymers are able to immobilize enzymes that produce reactions to substances) as well as to identify gases that may be harmful to the human body [6]. Another useful polymer subset is electrets. These polymers have the ability to store electrical charges thereby causing permanent polarization that can be used in the fabrication of structures that produce electric signals. These fabricated structures include films that are used in the construction of transducers. Additionally, ferroelectric polymers, poled by strong electric fields, can be used to create thermal sensors [10]. Smart polymers are extremely versatile and functionally malleable to fit the needs of various industries. In many cases, it is imperative to know when a certain temperature range is reached rather than an accurate measurement. Being able to quickly and visually assess the environmental conditions like temperature, is especially relevant in the food packaging and transportation industry as there is a need for a cheap and disposable sensor. For the benefit of a complete study and as well as from the perspective of the scientific community, the appropriate information is included. Although not entirely relevant to the current scope of the paper, food packaging sensors, that address other parameters of interest for monitoring the quality of food, are summarized. A thermochromic polymer sensor would be passive, i.e., it would not require a power source and with polymers being highly malleable materials, they can easily be extruded or 3D printed into a variety of shapes and sizes specific to their applications. Polymers are already known to be low cost and the effort of installation of a passive, self-regulating sensor would be much lower than an electric or electronic sensor. 

In the United States, 30–40% [13] of the food and food related products are discarded each year without consumption. This wastage of food occurs at both the retail and consumer levels for multiple reasons; one such reason is a lack of understanding of what shelf life really means. This is often due to flawed transportation, storage management as well as improper packaging. Consequently, the development of in-built monitoring systems for packaging to keep track of the quality of products during distribution is a topic of active research with the ultimate goals of improving consumer safety. Currently, temperatures inside a delivery vehicle cargo, packaging, warehouse, and showcases are assumed to be the desired temperature. However, sometimes there is a measurable discrepancy between the temperature of food and the surroundings, thus compromising the quality and shelf life of food and other transported goods. In order to address these issues, it is necessary to provide a bridging technology that is capable of gauging the food temperature from the monitored surrounding temperature, thereby alerting consumers and retailers of the quality of food and biologics [12].

A thermochromic temperature sensor is a device that can help to monitor temperature. It is a simple, inexpensive device that can show an easily assessable temperature change that reflects the conditions of a product to which it is attached. It provides a simple and efficient way to assess the quality of food with the naked eye. In this review, we focus on potential thermochromic polymer candidates for the development of an easily legible and minimally invasive thermal sensor. The end goal is to create an adaptable relatively low cost, easily fabricated, and easy to use temperature sensor that is compatible with food packaging criterion.

## 2. Prospective Polymer Candidates 

### 2.1. Polydiacetylene (PDA)

Conjugated polymers are materials which have the potential to be used in diverse applications in interdisciplinary areas due to their structural, spectral, and optical characteristics. A good example of this is polydiacetylene (PDA) The feature of interest in this study is their unique colorimetric and fluorescent transitions they undergo in response to an external stimuli (temperature) [14]. Unlike most of their other counterparts, polydiacetylene can undergo polymerization without the help of chemical initiators, catalysts and high temperatures. This is important as polymerization allows for simple processing of diacetylene [14]. However, this is not the most important aspect of PDAs. The absorption peak of approximately 640nm, due to electron delocalization, appears optically as a blue color. This blue shift is caused by the conformational change of the polymer backbone from planar to nonplanar [14]. The first known PDA based sensor was created in 1993 by Charych et al. [15]. They created a selective sensor for the influenza virus. However, PDAs are being used in a vast number of applications such as cell imaging, tumor targeting and now, in this study, as a thermal sensing system.

During a change in temperature, PDAs undergo a colorimetric transition from blue to red. Nevertheless, the most important aspect of PDAs is their reversibility within a certain temperature range. This property is usually achieved by strengthening their intermolecular interactions.

This study focuses on the use of thermochromic polymeric sensors in food packaging applications. An ideal candidate is a polymer which provides a clear visual color change without the need for wires or other complicated appendages that are usually attached to thermocouples for temperature monitoring. Consequently, PDA was one of the candidates that was considered in the temperature sensing system. This is because of the previous work done by Huo et al. which shows that PDAs can be modified to present a color change over a wide range of temperatures up to 250 °C while maintaining thermal stability. Huo et al. studied different compositions of PDAs. These PDAs (labelled 2a,2b,2c) were obtained via the Glaser reaction using a diacetylene moiety as a polymerizable core, amino acid moiety as a linking agent and two pyrene units as bilateral head group [16]. Figure 2 depicts the different PDAs (labelled 2a,2b,2c) that were made.

Thermogravimetric analysis was performed on the PDAs between 25 and 800 °C and they were observed to maintain their stability up to temperatures as high as 300 °C. The results can be seen in Figure 3.

Thermochromic Transitions in Polydiacetylene.

Thermochromism in PDAs was investigated by heating blue polymer films at 5 °C min^−1^ under vacuum and the visible color change observed is presented in Figure 4. Real time monitoring was performed using a fiber optic absorption spectrometer. It was observed (Figure 4) that the color change was reversible. For example, in PDA 2a, a shift in the original blue color at 25 °C to a red at 250 °C and a change back to blue when it is cooled to its original temperature is observed. The PDA 2b shows a similar change but the most distinct color change comes from the PDA 2c at 300 °C.

Yoon et al. also explored PDA. They developed diacetylene (DA) supramolecules that are inkjet printable which makes it highly advantageous in terms of ease of printability on large surfaces in a high throughput manner as well as its low production cost along with low waste materials produced [17]. Paper was used as the substrate which makes it suitable for not only scientific analysis but also in daily life applications [17]. A single component ink system that was compatible with the ink cartridge nozzle was created. It was composed of DA, bisurea and oligoethylene oxide moieties [17]. The DA monomers generated PDA when it was irradiated with ultraviolet rays. The other components enhanced supramolecular assembly of the monomers and increased water compatibility respectively [17].

DA 1-DA 3 were prepared by an increase in the number of ethylene oxide units. It increased from 4 (DA 1) to 7 (DA 2) and to ca. 10 (DA 3). Consequently, this causes a corresponding enhancement in the hydrophilic property of the monomer. However, in the case of DA 3, the exact number of the ethylene oxide group is unknown because it is prepared from commercially available poly (ethylene glycol) methyl ether only having an average molecular weight of 550 [17].

Three different samples with varying yields of DA were used: 77% of DA 1, 80% of DA 2, 70% of DA 3. The monomers were difficult to disperse in water; thus a maximum concentration of about 3 mM was used in this study. When the sample was irradiated, it changed colors from pale yellow to an intense blue. Samples were then printed out and placed on a hot plate to observe the color change which is displayed in Figure 5.

Both DA2 and DA3 displayed shifts in their absorption spectra with change in temperature. In both samples, the color changed from blue to purple at 80 °C and then to red at 120 °C. At ~200 °C, the color turned yellowish red and, above that, it was yellow. Once cooled to 30 °C, the color changed to red. DA3 had two temperature ranges in which it displayed reversible color change (30 to 80 °C and 170 to 30 °C). This method had the potential to be applicable to paper-based devices, temperature sensors and even anti-counterfeiting systems [17].

PDA infused films have also been used for bacterial [18] and fungal detection [19] in biosensing applications and have been used for rapid bacterial detection in food [20,21]. While these are not thermochromic properties, they are still desirable qualities for food packaging and make PDA a fitting candidate for a possible multipurpose sensor.

Currently, there is a patent for a diacetylene thermochromic ink that has been used in food packaging. The invention is composed of a bi-block temperature-sensitive color-changing ink and exhibits uniform color component dispersion, strong viscosity with printed matter, and no color drop phenomenon [22].

### 2.2. Polyaniline (PANI)

There is a need for materials that are cost effective and durable while also having ease of workability and good accessibility when it comes to packaging. Polyaniline (PANI) is a promising candidate with all these features. It exhibits optimal chromogenic properties in a wide range of temperatures while maintaining its structural integrity [23].

PANI has been studied extensively for multiple purposes. In 2002, Rannou et al. showed that diester doped PANI exhibited two glass transition temperatures with a strong thermochromic effect in UV−vis−NIR spectra at sub-zero temperatures. 4-sulfophthalic acid was used as the dopant to protonate PANI and lower the glass transition temperature which improved the flexibility of the films created. It is assumed that the thermochromic phenomenon is induced by temperature dependent torsions within the polymer chain. These torsions cause a widening of the band gap within the now non-planar geometry of the polymer chain which results in a hypsochromic shift of the band corresponding to the π−π* transition [8].

PANI was protonated in solution with several diesters at 0.5 wt% and at a ratio that ensured full protonation of the doped polymer. Films were then made on polypropylene and glass substrates at 45 °C. Its molecular structure is shown in Figure 6. The thermochromic effect was characterized by recording solid-state UV−vis−NIR spectra of the films and the glass transition temperatures were determined by DSC measurements.

The study concluded that films made of PANI protonated with 1,2-(di-2-(butoxyethyl)) ester of 4-sulfophthalic acid (DBEEPSA) exhibited the strongest thermochromic effect at temperatures below the glass transition temperatures of the diester dopants. The UV-vis-NIR spectra of PANI(DBEEPSA)0.5 demonstrates that, above the glass transition temperature, Tg (−55.15 °C or 218 K), there are minimal changes in absorption spectra, but at 6.85 °C (280 K), there is a sharp peak at 440 nm and a notably broad absorption tail which can be attributed to delocalization of charge carriers in PANI [14]. Another small peak is observed at ~ 800 nm which is indicative of localized charge carriers present within the system. Similar spectral changes were observed in PANI doped with 1,2-(di-2-ethylhexyl) ester of 4-sulfophthalic acid (DEHEPSA) but at lower temperatures of T < -65.15 °C (208 K) which corresponds to the lower T_g_ of the dopant [8]. The optical spectra of these samples are presented in Figure 7.

The US Food and Drug Administration, US FDA, recommends that cold foods be transported at 40 °F or below which corresponds to ~4.45 °C (277.6 K) [24]. PANI(DBEEPSA)0.5 is the ideal candidate for food packaging as it exhibits thermochromic behavior at 6.85 °C (280 K).

Previously, PANI has also been shown to be an efficient scavenger of free radicals such as 1,1-diphenyl-2-picrylhydrazyl (DPPH). Free radicals are a known cause of food spoilage; thus, this property of PANI can be implemented in the protection of consumable goods. The antioxidant properties of conducting polymers were first evaluated by Ismail et al. by determining the efficacy of PANI films that were used to protect vulcanized rubber from oxidation and radiation deterioration [25]. Though other polymers have been reported to display similar properties, PANI’s superior free radical scavenging ability is unmatched [26].

In 2011, Nand investigated polyaniline for packaging applications by determining the free radical capacity of heat-treated samples of polyaniline of granular and micro/nanorod morphologies [27]. 600 mg specimens were heated for 30 min at temperatures ranging from 100 to 300 °C in steps of 15 °C and cooled to room temperature. A radical scavenging assay was performed by measuring the absorbance of a sample treated with DPPH using a spectrophotometer and the amount of remaining DPPH was calculated. It was observed that all morphologies of PANI exhibited free radical scavenging properties at elevated temperatures, but this ability declined after thermal treatment above 200 °C. The samples were amorphous at these temperatures. and thus, have the potential to be successfully merged with commonly used thermoplastics for food packaging applications [26].

Detailed studies have shown that PANI is also resistant to a broad range of bacteria including Escherichia coli, Staphylococcus aureus, Pseudomonas aeruginosa, Enterococcus faecalis, and Campylobacter jejuni [28,29,30,31]. PANI has been incorporated into surfaces or used as surface coatings for infection control and food safety applications [32]. It is thought that the inherent electrical conductivity of the polymer interferes with the negatively charged bacterial cell surface through electrostatic adherence [28,29,33]. Robertson and group found that PANI also stimulates hydrogen peroxide based oxidative stress in bacterial cells which causes damage and potential cell death [34]. This is significant because it can be incorporated into packaging to protect items from spoilage due to contamination from pathogenic agents.

### 2.3. Polythiophenes (PT)

Polythiophenes are known for their high stability, ease of structural modification and controllable optical and electrochemical properties [35]. Most polythiophene derivatives exhibit various optical properties such as thermochromism, photochromism, etc. which make them excellent candidates for various advanced optoelectronic applications, one of them being as a thermochromic sensor. In these materials, the changes in color that are observed during temperature fluctuations are due to the conformational changes within the long backbone chains of conjugated polymers. The torsions and twists modify the conjugation length which in turn cause a shift in the absorption bands in the UV-vis-spectra which is observed as a color change. The alkoxy groups in thiophene rings at the 3-position contain an oxygen atom with lone pairs of electrons that conjugate with the polymer backbone which causes the conformational change and enhances its conductivity also [35].

Valderrama-Garcia et al. synthesized novel thiophene monomers containing pyrene units and alkyl spacers linked by an ester group called TPM1-5 which were used to synthesize polythiophenes (TPP1-5). TPM1 has a carbonyl group attached directly to the pyrene unit and TMP5 has it attached to the thiophene ring. The other monomers (TPM2-4) have the ester function located at various positions along the unit to determine the effect of the position of the carbonyl group on the optical and electronic properties [35]. These are then prepared into thin films over ITO/glass plates as substrates and the samples are excited at 360 nm in their solid state. The study concludes that all polymers exhibit an absorption band at 245 nm, determined by absorption and fluorescence spectroscopies, which is followed by a broad band at 350 nm [35]. This is shown in Figure 8. Table 1 presents a summary of the resulting optical properties.

In 2020, Mehta et al. reported the fabrication of reversible thermochromic films composed of Polyethylene dioxythiophene–polystyrene sulphonate (PEDOT:PSS) that were tuned with varying concentrations of copper chloride dihydrate to show thermochromic properties over a range of temperatures from 40 °C to 90 °C. The films were screen printed and examined using UV visible absorption spectroscopy, in situ vibration spectra and XRD spectra. The results indicated that an increased volume of concentrated copper chloride dihydrate and double-layer printing produces an observable color change at 65° C. They also developed a mobile app to analyze the RBG histogram of the captured images and match them to their corresponding temperatures thus setting them up for possible future commercial use [36]. The transmission spectra of these samples are presented in Figure 9. The resulting photographic images of the samples are shown in Figure 10.

Currently, there are not many polythiophene based sensors that are available commercially for the food industry. However, there is a patent under the Rhode Island Board of Education for thermochromic polymer-based temperature indicator consisting of a mix of polythiophene at varying weight percentages from 0.05 to 5.0% and a carrier medium. The device exhibits a color change when a set temperature or beyond is reached [37].

### 2.4. Polylactic Acid (PLA)

Most chromogenic materials are known to be toxic or carcinogenic and thus considered incompatible for use in in the medical or food industry. Polylactic acid (PLA) is non-toxic and biodegradable which makes it a viable candidate for use in sensors within the industry. However, PLA by itself does not have thermochromic abilities. In 2013, Seeboth et al. addressed this deficiency by creating a novel polymer system with PLA as the matrix polymer along with anthocyanidins as the thermochromic dye. Anthocyanidins are found in nature and are required for colors ranging from red to blue that are found in various fruits and flowers [38]. The molecular structures of PLA and anthocyanidins are presented in Figure 11. 

Rectangular polymer composite samples were created by extruding the composite and then using a laboratory press to create rectangular sheets of varying thickness (0,5, 1, 2 mm). Absorption spectra, presented in Figure 12, were measured over the range of 400 nm to 800 nm with a spectrometer [38].

The polymer composite was heated from 20–70 °C and it was determined to have two transitions. The first one was at temperatures in the range of 40–50 °C which corresponds to the glass transition of the polymer matrix followed by the melting of hexadecenoic acid rich regions in 60–70 °C range. During the first transition, there is a color change from red to violet due to structural changes in the cyanidin dye from its neutral state to its anionic form. The composite becomes clear during the second transition phase. This phenomenon shows promise in the development of new strategies for addressing design of temperature sensors without an external power source. Another advantage of this process is that it uses the commonly used and efficient method of extrusion to create samples [38]. A summary of various thermochromic polymers, their operational mechanism and their temperature sensitivity range are presented in Table 2. Table 3 highlights various sensing mechanisms and the feedback system that are used to identify spoilage in food packaging systems. A recurring concept in this Table is the color change mechanism triggered by not only gases but also temperature.

## 3. Conclusions

Thermal control and temperature monitoring are critical aspects of this field as any changes are indicative of the circumstances within a system and can imply the need for intervention. A misevaluation of the thermal conditions in the system can cause unwarranted product wastage. In this study, various investigations on the thermochromic properties of polymers have been reviewed to determine the best candidates for a temperature sensor that would be appropriate for food and biologics packaging and transport. Four candidates were considered: Polydiacetylene (PDA), Polyalanine (PANI), Polythiophenes and Polylactic acid (PLA).

PDA has many promising qualities making it an ideal candidate for use in thermochromic packaging. Their ability to be processed as DA super molecules allows them to be produced at low cost and they also display sensitivity to microbials which can alert the consumer if the product is contaminated. PANI is an attractive candidate as it not only has antimicrobial properties but shows thermochromic changes at ultra-low temperatures which would be ideal for vaccine transport and monitoring. Several of the new vaccines for COVID-19 are stored at extremely low temperatures of −80°C to −60 °C [49]. Polythiophenes are widely used in various thermal sensing tasks but the novelty in the most recent investigation is the fabrication of functional films through screen printing and using a mobile app in conjunction to determine the temperature. There are numerous future commercial prospects of this method as it reduces the cost of analysis and it is easy to use. It standardizes the color change assessments and makes it applicable to various fields. Furthermore, polythiophenes show color changes due to redox reactions of pyrene rings within the polymer chain which can be used for not only temperature sensing but also to detect environmental changes such as humidity, composition, and pH. Polylactic acid is completely non-toxic and thus can be used in food packaging without any concerns of contamination from chemical by-products.

In summary, it can be concluded that further development in the field of the application of thermochromic polymeric materials for temperature sensors is needed. The vast range of possible applications for chromogenic materials, especially polymers, leaves room to further explore more complex device mechanisms in the future. They have excellent prospects in the medical field that need not be confined to the cold chain monitoring processes. Calibration of color changes is an obstacle that only one study has investigated, and further research is required. Interactions between different chromogenic materials also need to be examined.

## Figures and Tables

**Figure 1 micromachines-12-01193-f001:**
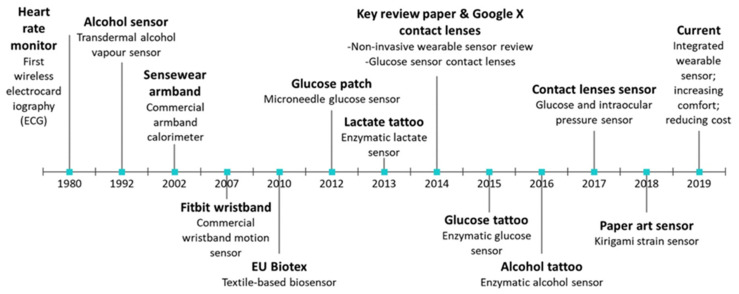
Key milestones of wearable sensors over the years. Reproduced with the permission of Reference [11]. Copyright 2021 Elsevier.

**Figure 2 micromachines-12-01193-f002:**
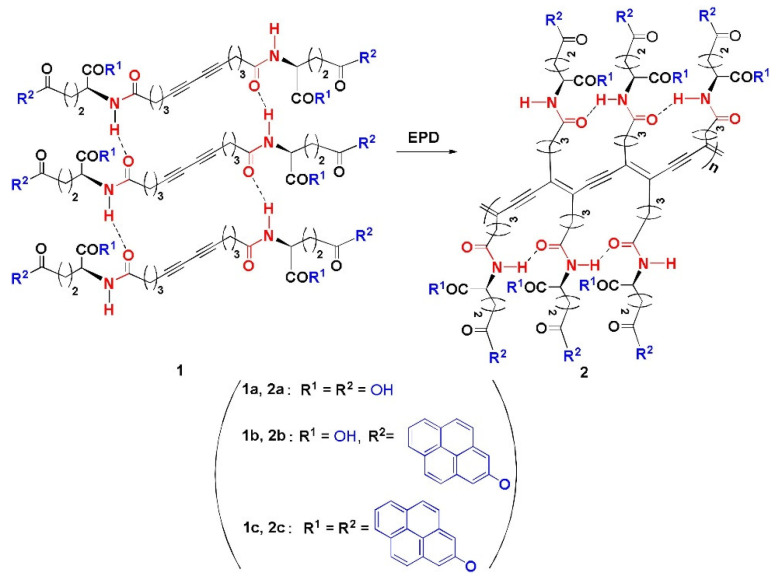
The EPD (electrophoteric deposition) process of polymeric products 2a–2c. Reproduced with the permission of Reference [16]. Copyright 2017, Elsevier.

**Figure 3 micromachines-12-01193-f003:**
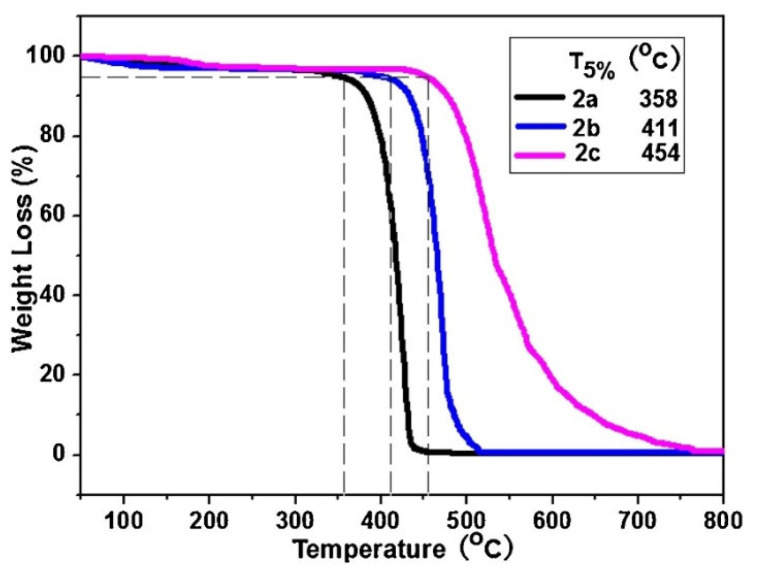
Thermogravimetric analysis of polymers, 2a–2c, collected under a N_2_ atmosphere. Reproduced with the permission of Reference [16]. Copyright 2017 Elsevier.

**Figure 4 micromachines-12-01193-f004:**
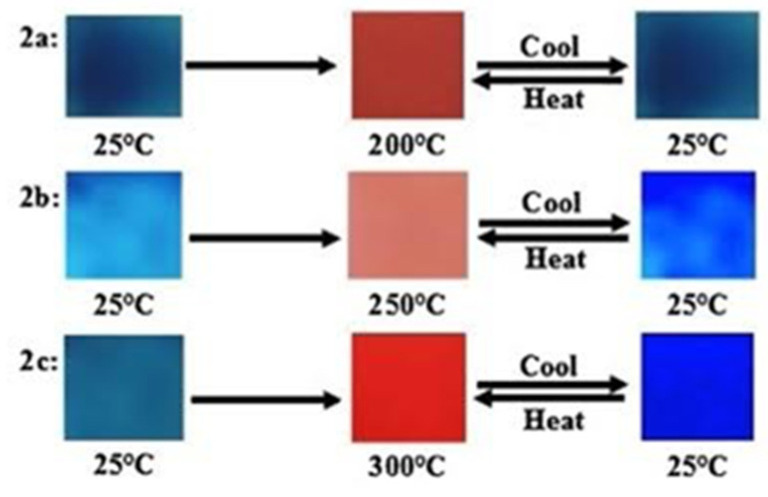
Photographs of the PDAs 2a–2c recorded upon increasing temperature from 25 °C to 300 °C and then cooling to room temperature. Reproduced with the permission of Reference [16] Copyright 2017 Elsevier.

**Figure 5 micromachines-12-01193-f005:**
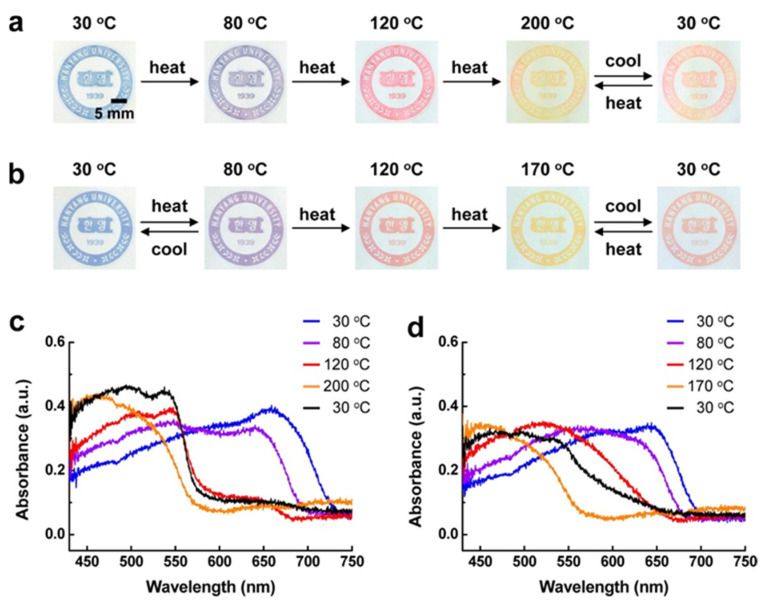
Photographs of printed images of (**a**) DA 2-derived PDAs and (**b**) DA 3-derived PDAs on an unmodified paper substrate upon thermal stimulation. UV–vis absorption spectra of (**c**) DA 2-derived PDAs and (**d**) DA 3-derived PDAs printed on an unmodified paper substrate as a function of the temperature. Black lines in each spectrum are the final phase involving cooling to 30 °C. Reproduced with the permission of Reference [17]. Copyright 2013 American Chemical Society.

**Figure 6 micromachines-12-01193-f006:**
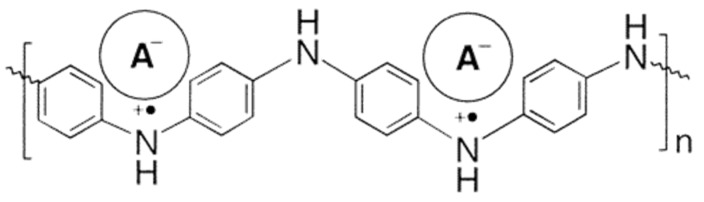
Molecular Structure of PANI(DEPSA)0.5. Reproduced with the permission of Reference [8]. Copyright 2002 American Chemical Society.

**Figure 7 micromachines-12-01193-f007:**
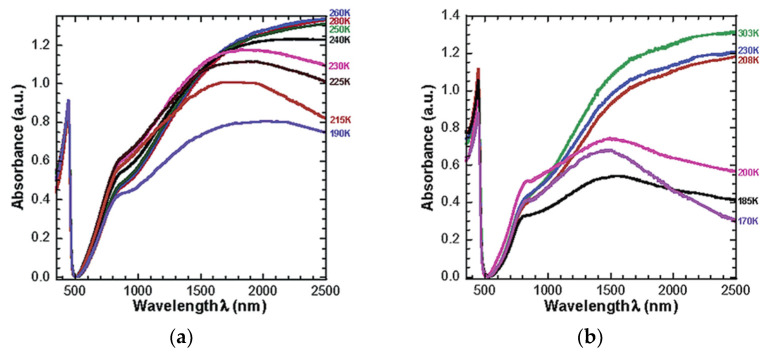
UV−vis−NIR spectra of (**a**) PANI(DBEEPSA)0.5 registered for the temperature range of 303−170 K (29.85 °C–103.15 °C). (**b**) PANI(DEHEPSA)0.5 registered for the temperature range of 280−190 K (6.85 °C–83.15 °C), Reproduced with the permission of Reference [8]. Copyright 2002 American Chemical Society.

**Figure 8 micromachines-12-01193-f008:**
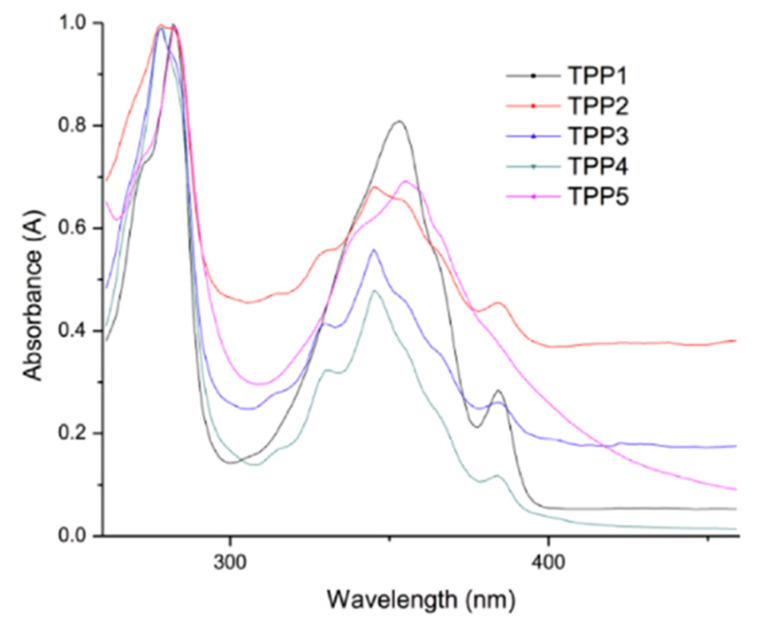
Normalized absorption spectra of the polymers in thin film deposited on ITO/glass. Reproduced with the permission of Reference [35]. Copyright Creative Common CC BY license.

**Figure 9 micromachines-12-01193-f009:**
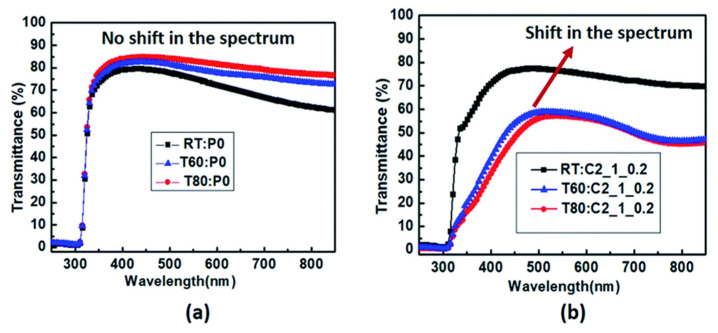
Transmission spectra for thin film samples of PEDOT:PSS at varying temperatures—room temperature (denoted as RT), 60 °C (denoted as T60) and 80 °C (denoted as T80) for (**a**) pristine PEDOT:PSS film (**b**) sample C2_1_0.2. Reproduced with the permission of Reference [36]. Copyright The Royal Society of Chemistry.

**Figure 10 micromachines-12-01193-f010:**
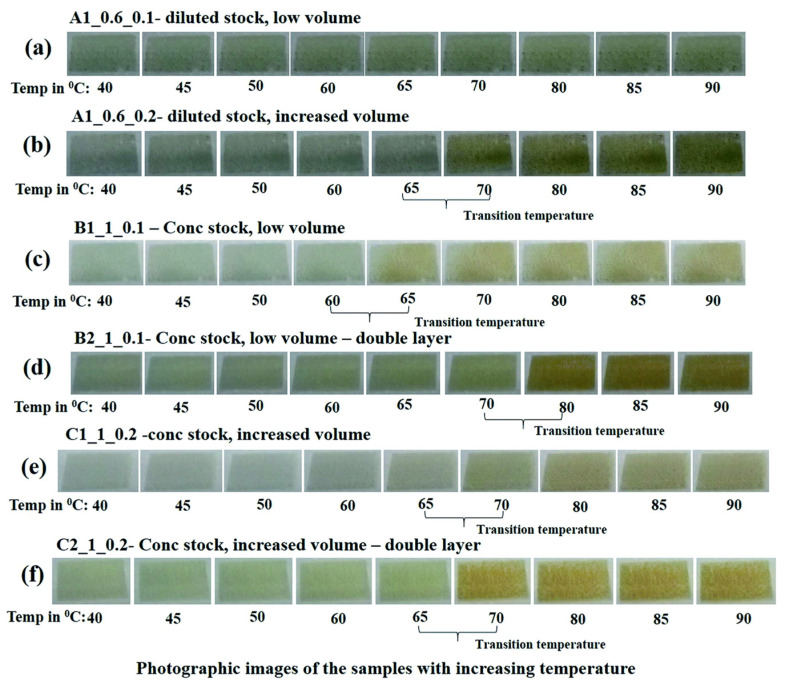
The photographic images of the samples (collected from the captured videos) heated at temperatures from 40–90 °C in steps of 5 °C. Reproduced with the permission of Reference [36]. Copyright The Royal Society of Chemistry.

**Figure 11 micromachines-12-01193-f011:**
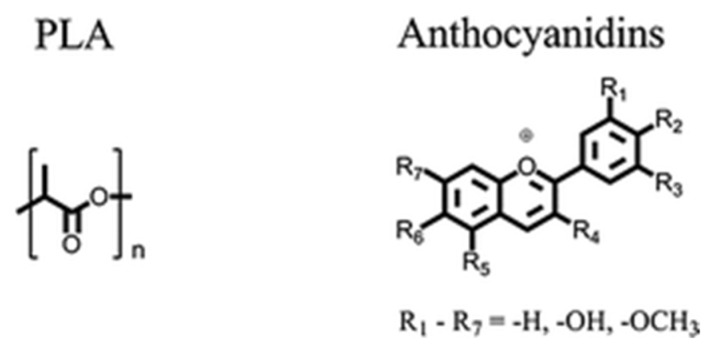
Molecular structures of PLA and anthocyanidins. Reproduced with the permission of Reference [39]. Copyright The Royal Society of Chemistry.

**Figure 12 micromachines-12-01193-f012:**
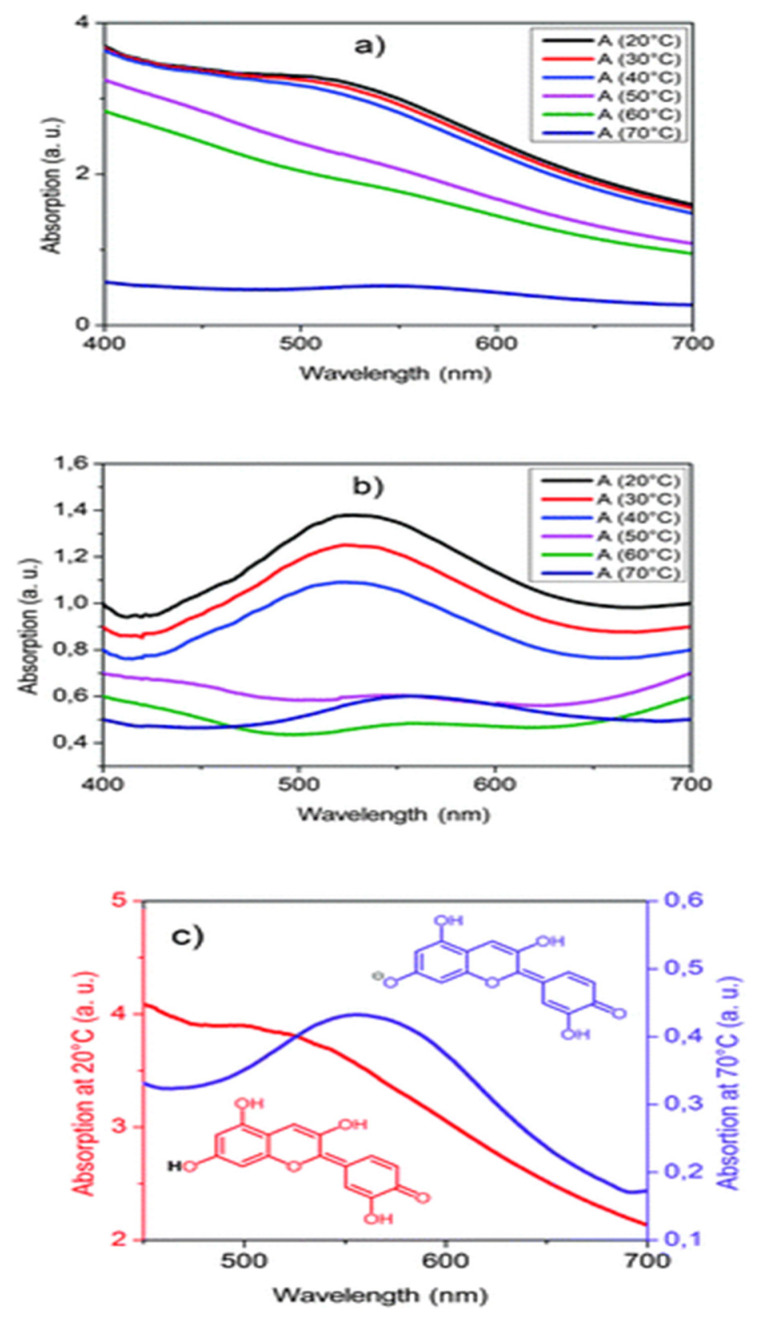
Temperature dependency of the visible absorption of PLA composite: (**a**) original curves on heating, (**b**) slope corrected curves on heating and (**c**) thermochromic switching effect, red curve: cold state, violet curve: hot state. Reproduced with the permission of Reference [38]. Copyright The Royal Society of Chemistry.

**Table 1 micromachines-12-01193-t001:** Optical Properties of polymers deposited on ITO/glass. Reproduced with the permission of Reference [35]. Copyright Creative Common CC BY license.

Compound	Absorption (λ nm)	Cut off (nm)	Emission (λ nm) λ_exc_ = 360 nm	Cut off (nm)
TPP1	275 ^b^–356 ^a^	450	388 ^c^, 409 ^c^, 545 ^d^	640
TPP2	273 ^b^–345 ^a^	450	391 ^c^, 410 ^c^, 539 ^d^	640
TPP3	273 ^b^–345 ^a^	450	389 ^c^, 409 ^c^, 543 ^d^	640
TPP4	274 ^b^–345 ^a^	450	397 ^c^, 413 ^c^, 543 ^d^	640
TPP5	275 ^b^–357 ^a^	450	390 ^c^, 409 ^c^, 544 ^d^	640

^a^: absorption band of the pyrene; ^b^: absorption band of the polythiophene backbone; ^c^: monomer emission; ^d^: excimer emission.

**Table 2 micromachines-12-01193-t002:** Summary of thermochromic polymeric materials and their properties.

Polymer	Operation Mechanism	Temperature Sensitivity Range	Color Change	Absorbance Wavelength	Ref.
Poly(lactic acid) (PLA) +Natural dye cyanidin chloride	Leuco Dye Development Solvent System	45 °C	Red Wine-Violet	530–560 nm	[39]
PMMA nanocomposite films+MMA organogels 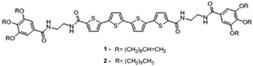	Conjugated Polymer	Up to 120 °C	Bright Green-Orange	575 nm	[20]
Poly{2,5-bis[3-(*N*,*N*-diethylamino)-1- oxapropyl]-1,4-phenylenevinylene} DAO-PPV + Toluene Solution	Conjugated Polymer	10–100 °C	Red–Yellow-Green	490–476 nm	[40]
Poly 2-methoxy-5-2 -ethylhexyloxy-1,4-phenylenevinylene PPV gel	Gel Film	25–100 °C	Red-Yellow	615 nm	[41]
Polyvinyl alcohol-borax-hydrogel (PVA)+Pyridinium N-phenolate betaine dye (DTPP)	Hydrogel-Indicator Dye System	5–80 °C	Colorless-Violet	550 nm	[42]
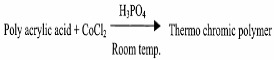	Gel Network–Inorganic Salt System	70–120 °C	Light Pink-Dark Blue	512–521 nm	[43]
Peptide-based amphiphilic polydiacetylene (PDA)	Conjugated Polymer	60–200 °C	Blue-Red	630–540 nm	[44]
Polyethylene Naphthalate (PEN) substrates coated with an indium tin oxide (ITO)		43–72 °C	Black-White		[45]
Nafion phenolphthalein crystal violet film	Polymer Matrix –Indicator Dye System	25–70 °C	Colorless-Pink	505 nm	[44]
Polyurethane	Leuco Dye based thermochromic pigment	38–60 °C	Light Red-White	440–590 nm	[46]
Poly(ethylene terephthalate) +Siloxane based elastomer	Cholesteric Liquid crystal	38 °C	Red shift	-	[47]
Poly(ethylene terephthalate) +Siloxane based elastomer	Cholesteric Liquid crystal	65 °C	Blue Shift	-	[47]

**Table 3 micromachines-12-01193-t003:** Developed sensors for monitoring of packaged food quality. Based on Reference [48].

Detection Target	Sensing Mechanism	Reporting Method	Substrate/Material	Development Stage
Time-temp	Redox reaction to oxygen defusion	Color change	Paper or plastic	Commercially Available
CO_2_	pH	Color change	Plastic film, aqueous solution	Proof of concept
	Luminescence dye	Fluorescence signal	Plastic film, organically modified silica matrix	Proof of concept
O_2_	Oxygen molecule quenches electronically excited lumophore	Luminescence	Silicone rubber, organic polymer	Proof of concept
	Redox dye, reducing agent, alkaline environment	Color change	Plastic, organic polymer	Commercially Available
Humidity	Change in the dielectric constant	Wireless detector device	Paper	Proof of concept
	Change in the dielectric constant	RFID tag wirelessly connected to a VNA	Paper	Proof of concept
	Change in the dielectric constant	RFID reader	Paper	Proof of concept
Chemical- TVBN	Chemosensitive Compounds	Color change	Polyaniline film	Proof of concept
Chemical- Array	Chemosensitive Compounds	Color change	Silica gel plate	Proof of concept
Bacteria	Antibody interaction	Bright-field imaging, electrochemical signal	Microfluidic device	Proof of concept
	Aggregation of paramagnetic silica beads correlates with the amount of DNA of interest	Cellular device	Paramagnetic silica beads	Proof of concept
	Graphene nanosheet traps E. coli	Change in impedance spectroscopy	Graphene based flexible acetate sheet	Proof of concept
	E. coli specific DNAzyme	Fluorescence	Cyclo-olefin polymer (COP)	Proof of concept

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
