# Peer review of "Thermochromic Polymeric Films for Applications in Active Intelligent Packaging—An Overview"

_micromachines, 2021, doi:10.3390/mi12101193_

Round 1
Reviewer 1 Report
The manuscript presents a very interesting review on the possible use of polymers for designing thermochromic plastic materials. However, it is neccessary to address to main aspects before publication:
-The introduction is not clear. It discusses so much aspects but the connection is not clear.
-Authors should include a section discussing the main conditions required for polymers to be used as thermocromic materials
Author Response
micromachines-1362495-Review 1:
The authors thank the reviewer for their very helpful comments and suggestions. The concerns are addressed below.
- The introduction is not clear. It discusses so much aspects but the connection is not clear.
We updated the introduction which is available below:
For the past decade, the scientific community has invested heavily into researching and identifying ways to map, monitor and control temperature in real-time due to the considerable influence temperature has on multiple aspects of our daily lives such as the life expectancy of diary, food and pharmaceuticals and in the manufacture, storage and transport of these products [1, 2]. Another critical example for the need to monitor and control temperature is in the storage, distribution and transportation of drugs, vaccines and biologics [3]. It has been reported that three of the COVID-19 vaccines that are most developed, i.e., Moderna, BioNTech and Pfizer need to be stored at extremely cold temperatures. Moderna’s COVID-19 candidate requires a storage temperature of -20°C while Pfizer’s vaccine requires ultra-cold temperatures, as cold as -70 °C [4, 5]. If these vaccines are exposed to temperatures that deviate even slightly from their ideal conditions, they may spoil and become unusable. Most vaccines, including the recently CDC approved Janssen Covid-19 vaccines by Johnson & Johnson, are usually stored between 2.22 °C and 7.78 °C. With the high demand for temperature measurements across multiple industries, various techniques and materials need to be considered to create sensors that are suitable for such diverse applications. One such technique that is examined in this review is the use of “smart” polymers in a sensing system to study, control and monitor the changes in temperature [6].
Smart materials are substances that react to a physical or chemical stimulus in its external environment with a quantifiable and reproducible response such as a change in material properties [6]. One such phenomenon is chromism, which is the ability of a substance to undergo color transition in response to a chemical or physical stimulus in its environment such as temperature, pH, humidity, pressure, etc. Smart polymers are increasingly being investigated in the last several decades. Polymers are adaptable to many applications due to their wide range of tunable chemical and physical properties which include structural stability, biocompatibility and ease of processing which makes them excellent candidates for subsequent integration with detection devices [6].
Thermochromism is the color change observed in chromic materials due to temperature fluctuations. Thermochromic systems are either direct or indirect. Direct systems change color with temperature differences in the environment, but indirect systems change colors when the temperature of the medium itself is affected. Stereoisomerism, liquid crystals, and molecular rearrangements are all examples of direct thermochromic systems while indirect systems need a combination of elements such as a leuco dye, a developer, and a solvent to form a functioning system [7]. Direct systems are advantageous as they are not dependent on additives for color transitions, which could be potentially toxic, and thus have a much more versatile field of applications. Temperature induced conformational changes within the polymer chain are responsible for the observed color changes due to causing a shift in the energy between the frontier orbitals. Changes in the structure leads to an increase in the energy gap, shorter conjugation lengths and wavelength shift in optical spectra [8].
Thermochromic materials have been used extensively in temperature sensing systems across various fields such as food packaging, medical thermography, the non-destructive testing of engineered articles, electronic circuitry, and pharmaceutical industry [9]. From a historic perspective, this year marks the 140th Birthday of Hermann Staudinger, the Father of Polymer Chemistry. This year is also the 101st anniversary of the proposal of the concept of macromolecules, that was characterized as polymers, by Staudinger in 1920 and the discovery of covalent bonds [10].
Sensors in general are often reflective of these diverse applications as scientists often need to control complex systems [11]. Figure 1 showcases the key milestones and progression of wearable sensors during the last 40 years. The first wearable sensor was a chest strap developed in 1980. This strap highlighted the importance of using flexible materials (such as polymers) as a substrate for the fabrication of sensors [12]. Polymers play an extremely vital role in these systems because they possess a variety of characteristics that enable them to adapt and meet the demand for a wide range of applications [13]. Essentially, polymeric materials can be tailored to respond in a favorable manner for whatever signal is required for control in sensing systems. For example, in health care, they can be used to mimic the natural behavior of sense organs (contact lenses), control of chemical reaction (modified polymers are able to immobilize enzymes that produce reactions to substances) as well as to identify gases that may be harmful to the human body [6]. Another useful polymer subset is electrets. These polymers have the ability to store electrical charges thereby causing permanent polarization that can be used in the fabrication of structures that produce electric signals. These fabricated structures include films that are used in the construction of transducers. Additionally, ferroelectric polymers, poled by strong electric fields, can be used to create thermal sensors [11]. Smart polymers are extremely versatile and functionally malleable to fit the needs of various industries. In many cases, it is imperative to know when a certain temperature range is reached rather than an accurate measurement. Being able to quicky and visually assess the environmental conditions like temperature, is especially relevant in the food packaging and transport industry as there is an urgent need for a cheap and disposable sensor. A thermochromic polymer sensor would be passive, i.e. it would not require a power source and with polymers being highly malleable materials, they can easily be extruded or 3D printed into a variety of shapes and sizes specific to its application. Polymers are already known to be low cost and the effort of installation of a passive, self-regulating sensor would be much lower than an electric sensor.
In the United States, 30-40% [14] of the food and food related products are discarded each year without consumption. This wastage of food occurs at both the retail and consumer levels for multiple reasons; one such reason is a lack of understanding of what shelf life really means. This is often due to flawed transportation, storage management as well as improper packaging. Consequently, the development of in-built monitoring systems for packaging to keep track of the quality of products during distribution is a topic of active research with the ultimate goals of improving consumer safety. Currently, temperatures inside a delivery vehicle cargo, packaging, warehouse, and showcases are assumed to be the desired temperature. However, sometimes there is a measurable discrepancy between the temperature of food and the surroundings, thus compromising the quality and shelf life of food and other transported goods. In order to address these issues, it is necessary to provide a bridging technology that is capable of gauging the food temperature from the monitored surrounding temperature, thereby alerting consumers and retailers of the quality of food [13].
A thermochromic temperature sensor is a device that can help to monitor temperature. It is a simple, inexpensive device that can show an easily assessable temperature change that reflects the conditions of a product to which it is attached. It provides a simple and efficient way to assess the quality of food with the naked eye. In this review, we focus on potential thermochromic polymer candidates for the development of an easily legible and minimally invasive thermal sensor. The end goal is to create an adaptable relatively low cost, easily fabricated, and easy to use temperature sensor that is compatible with food packaging criterion.
- Authors should include a section discussing the main conditions required for polymer.
We have added a small section on thermochromic mechanisms in long chain polymers and the reasons for picking a polymer system. The section is available below.
Thermochromism is the color change observed in chromic materials due to temperature fluctuations. Thermochromic systems are either direct or indirect. Direct systems change color with temperature differences in the environment, but indirect systems change colors when the temperature of the medium itself is affected. Stereoisomerism, liquid crystals, and molecular rearrangements are all examples of direct thermochromic systems while indirect systems need a combination of elements such as a leuco dye, a developer, and a solvent to form a functioning system [7]. Direct systems are advantageous as they are not dependent on additives for color transitions, which could be potentially toxic, and thus have a much more versatile field of applications. Temperature induced conformational changes within the polymer chain are responsible for the observed color changes due to causing a shift in the energy between the frontier orbitals. Changes in the structure leads to an increase in the energy gap, shorter conjugation lengths and wavelength shift in optical spectra [8].
Smart polymers are extremely versatile and functionally malleable to fit the needs of various industries. In many cases, it is imperative to know when a certain temperature range is reached rather than an accurate measurement. Being able to quicky and visually assess the environmental conditions like temperature, is especially relevant in the food packaging and transport industry as there is an urgent need for a cheap and disposable sensor. A thermochromic polymer sensor would be passive, i.e. it would not require a power source and with polymers being highly malleable materials, they can easily be extruded or 3D printed into a variety of shapes and sizes specific to its application. Polymers are already known to be low cost and the effort of installation of a passive, self-regulating sensor would be much lower than an electric sensor.

Reviewer 2 Report
This review article reports the thermochromic property for sensing mechanism in food packaging related applications using polymeric materials. It is useful for technological application and further research. The manuscript has been written short (but nice) and requires additional modifications. It seems that sample details (condition, concentration etc) need more information (it was used directly from the references) for understanding the results. Several of the comments are given below which can be useful to revise.
1.Page 2: …30-40-% discard of food …add a reference.
2.The introduction part seems more common about the importance of food packaging. Detail more about the importance or role of thermochromic polymer materials, operation mechanism, its practical cost, sensitivity, selectivity etc.
3.In fig. 2, provide the additional information about the change in the reaction since 1a-c and 2a-c could not be see in the picture and did not defined the sample condition. Provide the full form of EPD.
4.In fig.5, what are DA 1-3?. Mention briefly about sample nature and explain.
5.Page 7, para 3: sulfothalic acid …spell error. Also in para 4, mention the glass transition temperature of PANI.
6.In fig. 7, whether the samples were treated at different temperature and measured the UV-Vis spectra or it was taken based on an in situ arrangement?. Also, mention the temperature in °C instead of °F and K scales in order to maintain uniformity as mentioned in PDA.
7.Page 8 last para: mention the different kinds of morphology.
8.Page 9 para 2: remove drug delivery system since the whole paragraph was about antimicrobial effect.
9.Page 9 :What are TPM 1-5, is there any compositional changes? Also in fig. 10 mention about the sample condition such as A-C. Application of thiophenes for food packaging was not given.
10.Page 13 last para: properties is presented in table…
11.Mention additional details with the practical application report of thermochromic polymers if available commercially or literature for food safety utilization.
12.Table 3: whether it is based on the discussed polymer or using other polymers?. If it is related to the discussed thermochromic polymers it is an added advantage. Or if it is about other polymers then it is out of scope of the present work. Also, gas and humidity based sensors were not discussed enough in the manuscript text.
- It would be better if the authors report several of their works and provide view point to advance further. It would support the summary in the conclusion section too.
Author Response
micromachines-1362495 -Review 2
The authors thank the reviewer for their very helpful comments and suggestions. The concerns are addressed below.
- 30-40% wastage of food … add a reference:
A Reference has been added – Reference [14]; Food Waste FAQs. 2021; Available from: https://www.usda.gov/foodwaste/faqs , accessed on 2nd Sep, 2021
- The introduction part seems more common about the importance of food packaging. Detail more about the importance or role of thermochromic polymer materials, operations mechanism, practical cost, sensitivity, selectivity, etc.:
The Introduction has been enhanced significantly. We have addressed sensitivity of temperature range in Table 3.
For the past decade, the scientific community has invested heavily into researching and identifying ways to map, monitor and control temperature in real-time due to the considerable influence temperature has on multiple aspects of our daily lives such as the life expectancy of diary, food and pharmaceuticals and in the manufacture, storage and transport of these products [1, 2]. Another critical example for the need to monitor and control temperature is in the storage, distribution and transportation of drugs, vaccines and biologics [3]. It has been reported that three of the COVID-19 vaccines that are most developed, i.e., Moderna, BioNTech and Pfizer need to be stored at extremely cold temperatures. Moderna’s COVID-19 candidate requires a storage temperature of -20°C while Pfizer’s vaccine requires ultra-cold temperatures, as cold as -70 °C [4, 5]. If these vaccines are exposed to temperatures that deviate even slightly from their ideal conditions, they may spoil and become unusable. Most vaccines, including the recently CDC approved Janssen Covid-19 vaccines by Johnson & Johnson, are usually stored between 2.22 °C and 7.78 °C. With the high demand for temperature measurements across multiple industries, various techniques and materials need to be considered to create sensors that are suitable for such diverse applications. One such technique that is examined in this review is the use of “smart” polymers in a sensing system to study, control and monitor the changes in temperature [6].
Smart materials are substances that react to a physical or chemical stimulus in its external environment with a quantifiable and reproducible response such as a change in material properties [6]. One such phenomenon is chromism, which is the ability of a substance to undergo color transition in response to a chemical or physical stimulus in its environment such as temperature, pH, humidity, pressure, etc. Smart polymers are increasingly being investigated in the last several decades. Polymers are adaptable to many applications due to their wide range of tunable chemical and physical properties which include structural stability, biocompatibility and ease of processing which makes them excellent candidates for subsequent integration with detection devices [6].
Thermochromism is the color change observed in chromic materials due to temperature fluctuations. Thermochromic systems are either direct or indirect. Direct systems change color with temperature differences in the environment, but indirect systems change colors when the temperature of the medium itself is affected. Stereoisomerism, liquid crystals, and molecular rearrangements are all examples of direct thermochromic systems while indirect systems need a combination of elements such as a leuco dye, a developer, and a solvent to form a functioning system [7]. Direct systems are advantageous as they are not dependent on additives for color transitions, which could be potentially toxic, and thus have a much more versatile field of applications. Temperature induced conformational changes within the polymer chain are responsible for the observed color changes due to causing a shift in the energy between the frontier orbitals. Changes in the structure leads to an increase in the energy gap, shorter conjugation lengths and wavelength shift in optical spectra [8].
Thermochromic materials have been used extensively in temperature sensing systems across various fields such as food packaging, medical thermography, the non-destructive testing of engineered articles, electronic circuitry, and pharmaceutical industry [9]. From a historic perspective, this year marks the 140th Birthday of Hermann Staudinger, the Father of Polymer Chemistry. This year is also the 101st anniversary of the proposal of the concept of macromolecules, that was characterized as polymers, by Staudinger in 1920 and the discovery of covalent bonds [10].
Sensors in general are often reflective of these diverse applications as scientists often need to control complex systems [11]. Figure 1 showcases the key milestones and progression of wearable sensors during the last 40 years. The first wearable sensor was a chest strap developed in 1980. This strap highlighted the importance of using flexible materials (such as polymers) as a substrate for the fabrication of sensors [12]. Polymers play an extremely vital role in these systems because they possess a variety of characteristics that enable them to adapt and meet the demand for a wide range of applications [13]. Essentially, polymeric materials can be tailored to respond in a favorable manner for whatever signal is required for control in sensing systems. For example, in health care, they can be used to mimic the natural behavior of sense organs (contact lenses), control of chemical reaction (modified polymers are able to immobilize enzymes that produce reactions to substances) as well as to identify gases that may be harmful to the human body [6]. Another useful polymer subset is electrets. These polymers have the ability to store electrical charges thereby causing permanent polarization that can be used in the fabrication of structures that produce electric signals. These fabricated structures include films that are used in the construction of transducers. Additionally, ferroelectric polymers, poled by strong electric fields, can be used to create thermal sensors [11]. Smart polymers are extremely versatile and functionally malleable to fit the needs of various industries. In many cases, it is imperative to know when a certain temperature range is reached rather than an accurate measurement. Being able to quicky and visually assess the environmental conditions like temperature, is especially relevant in the food packaging and transport industry as there is an urgent need for a cheap and disposable sensor. For the benefit of a complete study and as well as from the perspective of the scientific community, Table 3 is included. Although not entirely relevant to the current scope of the paper, it summarizes food packaging sensors that address other parameters of interest for monitoring the quality of food. A thermochromic polymer sensor would be passive, i.e. it would not require a power source and with polymers being highly malleable materials, they can easily be extruded or 3D printed into a variety of shapes and sizes specific to its application. Polymers are already known to be low cost and the effort of installation of a passive, self-regulating sensor would be much lower than an electric sensor.
In the United States, 30-40% [14] of the food and food related products are discarded each year without consumption. This wastage of food occurs at both the retail and consumer levels for multiple reasons; one such reason is a lack of understanding of what shelf life really means. This is often due to flawed transportation, storage management as well as improper packaging. Consequently, the development of in-built monitoring systems for packaging to keep track of the quality of products during distribution is a topic of active research with the ultimate goals of improving consumer safety. Currently, temperatures inside a delivery vehicle cargo, packaging, warehouse, and showcases are assumed to be the desired temperature. However, sometimes there is a measurable discrepancy between the temperature of food and the surroundings, thus compromising the quality and shelf life of food and other transported goods. In order to address these issues, it is necessary to provide a bridging technology that is capable of gauging the food temperature from the monitored surrounding temperature, thereby alerting consumers and retailers of the quality of food [13].
A thermochromic temperature sensor is a device that can help to monitor temperature. It is a simple, inexpensive device that can show an easily assessable temperature change that reflects the conditions of a product to which it is attached. It provides a simple and efficient way to assess the quality of food with the naked eye. In this review, we focus on potential thermochromic polymer candidates for the development of an easily legible and minimally invasive thermal sensor. The end goal is to create an adaptable relatively low cost, easily fabricated, and easy to use temperature sensor that is compatible with food packaging criterion.
- In Fig 2, provide the additional information about the change in the reaction since 1a-c and 2a-c could not be seen in the picture and did not define the sample condition. Provide the full form of EPD:
These PDAs (labelled 2a,2b,2c) were obtained via the Glaser reaction using a diacetylene moiety as a polymerizable core, amino acid moiety as a linking agent and two pyrene units as bilateral head group. EPD is short for electrophoteric deposition.
- In Fig 5, what is DA 1-3? Mention briefly about the sample nature and explain:
DA 1-3 are diacetylene (DA) supramolecules that are inkjet printable. DA 1-DA 3 were prepared by an increase in the number of ethylene oxide units. It increased from 4 (DA 1) to 7 (DA 2) and to ca. 10 (DA 3). Consequently, causing a corresponding enhancement in the hydrophilic property of the monomer. However, in the case of DA 3, the exact number of the ethylene oxide group is unknown because it is prepared from commercially available poly (ethylene glycol) methyl ether only having an average molecular weight of 550 [18].
- Page 7, para 3: Sulfothalic acid spell error. Also in para 4, mention glass transition temperature of PANI:
We fixed the spelling error and mentioned the glass transition temperature of PANI which is (-55.15 °C or 218 K).
- In Fig 7, whether samples were treated at different temperatures and measured the UV-Vis spectra or it was taken based on an in situ arrangement. Also mention the temperature in C instead of F and K scales to maintain uniformity as mentioned in PDA:
The samples were measured using UV−vis−NIR spectra of a) PANI(DBEEPSA)0.5 registered for the temperature range of 303−170 K (29.85 °C - -103.15 °C). b) PANI(DEHEPSA)0.5 registered for the temperature range of 280−190 K (6.85 °C - -83.15 °C). All K and F values were converted to Celsius.
- Page 8, last para; mention the different morphologies:
granular and micro/nanorod morphologies
- Page 9 para 2; Remove drug delivery system since the whole paragraph was about antimicrobial effect:
We did not remove the antimicrobial properties of PANI because we think it is relevant in the food packaging industry. We have provided more resources which deal with usage of PANI as an antimicrobial agent in food packaging.
- What are TPM 1-5, is there any compositional changes? Also in fig 10 mention about the sample condition such as A-C. Application of thiophenes for food packaging was not given:
TPM 1-5: TPM1 has a carbonyl group attached directly to the pyrene unit and TMP5 has it attached to the thiophene ring. The other monomers (TPM2-4) have the ester function located at various positions along the unit to determine the effect of the position of the carbonyl group on the optical and electronic properties. In fig 10, the photographic images of the samples (collected from the captured videos) heated at temperatures from 40–90 °C in steps of 5 °C. We added a reference for a patented polythiophene polymer sensor that is available for food packaging.
- Mention additional details with the practical application report of thermochromic polymers if available commercially or literature for food safety utilization:
Table 3 addresses current developed sensors used for monitoring packaged food quality.
- Table 3: whether it is based on the discussed polymer or using other polymers? If it is related to the discussed thermochromic polymers, it is an added advantage. Or if it is about the other polymers then it is out of the scope of the present work. Also, gas and humidity based sensors were not discussed enough in the manuscript text:
For the benefit of a complete study and as well as from the perspective of the scientific community, Table 3 is included. Although not entirely relevant to the current scope of the paper, it summarizes food packaging sensors that address other parameters of interest for monitoring the quality of food
- It would be better if the authors report several of their works and provide view point to advance further. It would support the summary in the conclusion section too:
Historically, the research group’s focus, at NJIT, has been on inorganic temperature measurement systems such as vanadium oxide-based microbolometers, amorphous silicon microbolometers, HgCdTe infrared detectors and silicides such as PtSi & RhSi on silicon Schottky barrier detectors. The shift in focus has been largely due to the changing circumstances surrounding biologics transport specifically regarding Covid 19 vaccine. We have recently moved towards 3D printable devices and thus we are currently focusing on polymer-based temperature sensors. The end goal of the study is to create a 3D printable polymer temperature sensor that allows us to instantaneously visually assess the temperature range of a system and has the potential to be commercially viable.
Reviewer 3 Report
This paper reviewed possible polymer candidates for use as temperature sensors in packaging for monitoring purpose. The authors listed ample information but the foundation of the argument is flawed.
The purpose of real-time monitoring is to record the temperature history of the measurand without loosing information, which is especially true for packaging because it is the thermal history that matters the most. Using thermochromic sensors can either only tell the instantaneous temperature if the sensor is reversible or the maximum or lowest temperature if the sensor is irreversible. Lost of critical information, e.g. duration at maximum temperature, cannot be avoided unless a separate monitoring device is used to constantly record the color of the package which defeats the purpose of the whole idea.
There are benefits such as convenient to read temperature but real-time monitoring is a wrong application.
Author Response
micromachines-1362495-Review 3
The authors thank the reviewer for their very helpful comments and suggestions. The concerns are addressed below.
This paper reviewed possible polymer candidates for use as temperature sensors in packaging for monitoring purpose. The authors listed ample information but the foundation of the argument is flawed.
The purpose of real-time monitoring is to record the temperature history of the measurand without losing information, which is especially true for packaging because it is the thermal history that matters the most. Using thermochromic sensors can either only tell the instantaneous temperature if the sensor is reversible or the maximum or lowest temperature if the sensor is irreversible. Lost of critical information, e.g. duration at maximum temperature, cannot be avoided unless a separate monitoring device is used to constantly record the color of the package which defeats the purpose of the whole idea.
There are benefits such as convenient to read temperature but real-time monitoring is a wrong application.
In many cases, it is imperative to know when a certain temperature range is reached rather than an accurate measurement. Being able to quicky and visually assess the environmental conditions like temperature, is especially relevant in the food packaging and transport industry as there is an urgent need for a cheap and disposable sensor. A thermochromic polymer sensor would be passive, i.e. it would not require a power source and with polymers being highly malleable materials, they can easily be extruded or 3D printed into a variety of shapes and sizes specific to its application. Polymers are already known to be low cost and the effort of installation of a passive, self-regulating sensor would be much lower than an electric sensor. In this review, we focus on potential thermochromic polymer candidates for the development of an easily legible and minimally invasive thermal sensor. The end goal is to create an adaptable relatively low cost, easily fabricated, and easy to use temperature sensor that is compatible with food packaging criterion.

Round 2
Reviewer 1 Report
Authors have addressed all my comments and now the manuscript is fully publishable.
Author Response
Thank reviewers to review this manuscript.
Reviewer 2 Report
The authors have answered to the comments raised by this reviewer and incorporated in the manuscript. However some clarifications should be pointed out. It can be accepted after the corrections given below.
- Comment no.8 was about the removal of drug delivery system and not about the antimicrobial properties for the food safety. It seems the comment was misunderstood.
- In conclusion para 1 starting with historically… should be placed in the introduction, not in the conclusion section.
Author Response
Dear reviewer,
We have revised this manuscript according to your comments, please see the highlights in the attachment.

Reviewer 3 Report
The authors' reply does not appear to address the critical flaw that was pointed out. Again, the reviewed sensors are not for real-time monitoring but for ease of inspection by human users. The sensors are not real time.
Author Response
Dear reviewer,
Thank you for your suggestions.